# Disentangling snakebite dynamics in Colombia: How does rainfall and temperature drive snakebite temporal patterns?

**Carlos Bravo-Vega**⊙*⊚, **Mauricio Santos-Vega**⊙⊚, **Juan Manuel Cordovez**⊙

Grupo de Investigación en Biología Matemática y Computacional (BIOMAC), Departamento de Ingeniería Biomédica, Universidad de los Andes, Bogotá, Colombia

⊚ These authors contributed equally to this work.
* ca.bravo955@uniandes.edu.co

**Data Availability Statement:** Data and codes to replicate the results of this study are going to be available in figshare server: DOI: 10.6084/m9.

## Abstract

The role of climate driving zoonotic diseases' population dynamics has typically been addressed via retrospective analyses of national aggregated incidence records. A central question in epidemiology has been whether seasonal and interannual cycles are driven by climate variation or generated by socioeconomic factors. Here, we use compartmental models to quantify the role of rainfall and temperature in the dynamics of snakebite, which is one of the primary neglected tropical diseases. We took advantage of space-time datasets of snakebite incidence, rainfall, and temperature for Colombia and combined it with stochastic compartmental models and iterated filtering methods to show the role of rainfall-driven seasonality modulating the encounter frequency with venomous snakes. Then we identified six zones with different rainfall patterns to demonstrate that the relationship between rainfall and snakebite incidence was heterogeneous in space. We show that rainfall only drives snakebite incidence in regions with marked dry seasons, where rainfall becomes the limiting resource, while temperature does not modulate snakebite incidence. In addition, the encounter frequency differs between regions, and it is higher in regions where *Bothrops atrox* can be found. Our results show how the heterogeneous spatial distribution of snakebite risk seasonality in the country may be related to important traits of venomous snakes' natural history.

## Author summary

Snakebite envenoming is a neglected tropical disease characterized by its high burden on the rural population and high mortality if antivenom is not administered. The ecology of this health problem is not well-understood; however, approaches to address the temporal are growing. So far, we know that rainfall can play an important role in driving snakebite incidence seasonality at a national scale. Moreover, geographical areas with high rainfall are more prone to have high snakebite risk, but the spatial heterogeneity of the temporal association (i.e., if there are different seasonal patterns of rainfall-incidence association in different geographical areas of a country) is just starting to emerge in the literature. By

figshare.15029700, (https://figshare.com/s/
305c1d006d433debe14d).

**Funding:** This study was partially supported by:
Minciencias, Colombia (https://www.minciencias.
gov.co/), Application 727 for doctoral student to
CBV, and Universidad de los Andes, Colombia
(https://uniandes.edu.co/), Funding program for
doctoral students, awarded to CBV. The funders
had no role in study design, data collection and
analysis, decision to publish, or preparation of the
manuscript.

**Competing interests:** The authors have declared
that no competing interests exist.

formulating and fitting compartmental models to data, we generated a flexible framework
that relies on temporal resolved datasets and a compartmental mathematical model to
understand the effect of climatic covariates (such as rainfall and temperature) driving
snakebite dynamics in space and time. We applied this framework to Colombia and
found that dry seasons cause a decrease in snakebite incidence: Rainfall only drives snake-
bite dynamics in regions with marked dry seasons. Thus, rainfall is a limiting resource of
the system, and its effect is not spatially homogeneous. On the other hand, the tempera-
ture had no significant effect driving snakebite incidence. Our modeling approach can
also be used to estimate the effect of climate anomalies on snakebite incidence and has the
potential to be used as a tool to monitor snakebite incidence.

## Introduction

Snakebite envenoming is the neglected tropical disease (NTD) with the highest mortality rate,
but despite its importance, data collection is challenging, and the actual disease burden is
underestimated [1,2]. The most reliable estimates are at least 1.8 million cases with 435.000
deaths each year. However, this data can underestimate the real burden because patients prefer
seeking traditional medicine instead of allopathic medicine, and mortality data do not quantify
the morbidity associated with physical and mental disabilities caused by some snakebites [3–
9]. Estimating the real burden is crucial to understand the ecological characteristics that modu-
late human-snake encounters, where the climate can play an important role in driving the
ecology of venomous snakes [10–13]. Although several studies have estimated snakebite risk's
spatial and temporal heterogeneity based on statistical analyses and climatic variables, the
nature of these models is empiric: Their main findings are important correlations between
snakebite risk and covariates [7,14–17]. Thus, their usage as disease monitoring tools is limited
due to their correlational nature, which mostly hides reality's complexity [1,18,19].

Compartmental modeling has emerged as a tool to understand the processes underlying
disease transmission, and it has been used widely for several NTDs [20–25]. This modeling
approach subdivides population into a set of compartments, where flows represent movement
between the compartments. Usually, these flows are important parameters, e.g., incidence,
mortality, and recovery rate. Thus, covariates can affect specific flows in the model, allowing
us to understand the relationship between covariates and disease spread and making the
model flexible and an excellent tool for generating fundamental knowledge about the basic
mechanisms behind disease spread [26]. Contrarily, empirical and correlational approaches
use general equations and parameters not based on disease dynamics, contributing with a
more general knowledge about the general behavior of disease spread [20,26]. Even so, com-
partmental models rely strongly on disease' ecology knowledge, and as in empirical models,
the quality of the data. For example, although snakebite envenoming is not an infectious dis-
ease, it is caused by the interaction between humans and venomous snakes, and that interac-
tion can be modeled similarly to an infectious disease. A previous study demonstrated that the
main assumption of several compartmental models (the law of mass action) could explain
snakebite geographical variation in Costa Rica [27]. By combining compartmental models
with spatial and temporal surveillance data, it is possible to have reliable estimations for multi-
ple epidemiological parameters and to perform fine political scale extrapolations of the model
on other countries with different covariates, making these models a valuable tool to under-
stand, manage and control several NTDs [28,29].

To apply compartmental modeling to snakebites, it is essential to understand the most important venomous snake species ' biology, which is often poor [11,30]. In the Neotropics, one of the most important venomous snake groups is the genus *Bothrops* (Wagler, 1824), which is distributed broadly from southern Mexico to northern Argentina and causes the majority of envenomings in this area [31–33]. This genus is viviparous with an average clutch size between 3.5–37 offsprings, mainly during the rainy season [10,12,34–36]. This seasonal dynamic usually increases the venomous snake population during the rainy season, thus increasing snakebite risk during this period. Additionally, rainy seasons can cause flooding, habitat perturbation, and increased prey abundance, making venomous snakes more active, resulting in an additional increase in snakebite risk [37–39]. Previous studies have found that snakebite temporal dynamics tend to be related to seasonal rainfall patterns in tropical countries such as Costa Rica and Sri Lanka, but these studies did not account for climatic heterogeneity [15,16]. Later, in Sri Lanka a study used multivariate Poisson process modeling to evaluate the spatio-temporal association between incidence and climatic and socioeconomic variables, thus finding the spatio-temporal apparition of hotspots in that country [40]. Therefore, the spatial heterogeneity of the temporal association between climate and snakebite incidence has been addressed by statistical modeling rather than more robust modeling approaches such as compartmental modeling, and it has not been studied in neotropical countries.

Colombia is an ideal setting to study this spatial heterogeneity given its location with diverse climatic conditions, where different regions have different rainfall and temperature patterns [41]. In the country, snakebite is a severe public health problem, where recent reports account for around 4500 envenoming cases with at least 40 deaths each year, and two species (*Bothrops asper* and *Bothrops atrox*) cause most envenoming's [42,43]. In this study, we developed and calibrated a compartmental model that can disentangle the role of rainfall and temperature on snakebite temporal patterns. We used compartmental stochastic models combined with iterated filtering statistical inference methods to explore the role of these climatic drivers on snakebite temporal patterns and to understand this association's geographic distribution.

## Materials and methods

We followed four steps to develop and fit a compartmental modeling approach that accounts for the role of temperature and rainfall on snakebite dynamics. *i)* We developed two compartmental models for national data to determine the effect of temperature and rainfall driving national snakebite incidence. *ii)* We divided the country into six regions with similar rainfall patterns: This had the aim to aggregate municipality data within these regions to reduce noise caused by the low occurrence of snakebites and heterogeneous climate patterns. *iii)* We proposed a modeling scheme with different compartmental models to test hypotheses for the association between rainfall and snakebite incidence for the country's six regions. *iv)* We fitted our modeling scheme in each region using an iterated filtering approach that maximizes likelihood [44].

### Climate and incidence data

**Incidence data.** We got publicly available snakebite incidence data between January of 2010 to the end of October of 2016 from the Sistema Nacional de Vigilancia en Salud Pública (SIVIGILA) of Colombia. This dataset is reported in epidemiological weeks, and it details the number of cases for each municipality (the smallest political unit in the country). We only worked with municipalities that reported snakebite for all the study years. First, we looked at

the epidemiological calendar of Colombia to convert the timescale of the reported cases from epidemiological weeks to months. We aggregated the weekly reported cases for each month. Then, for weeks that overlapped between two months, we distributed the cases by weighting them based on the number of days of the week that belong to each month. Finally, we removed the temporal trend of the timeseries using a locally estimated scatterplot smoothing ('stats' package, R environment v. 4.1.1. [45,46]) to only account for seasonality and inter-annual variation in the data.

**Rainfall and temperature data.**   We got monthly raster maps for minimum and maximum temperature and rainfall for Colombia between the year 2010 to the end of October 2016 with a resolution of ~ 21.23 km$^2$ from TerraClimate dataset (http://www.climatologylab.org/terraclimate.html) [47]. In addition, we removed climate data from areas where the two most important venomous snakes' species in the country are absent by using an altitude threshold of 1900 m.a.s.l. for *Bothrops asper* and of 1500 m.a.s.l. for *Bothrops atrox*, which are the species that cause most envenomings in the country [33].

**Clustering of regions with similar rainfall pattern.**   We used a clustering algorithm over rainfall maps to define regions with similar rainfall patterns. Given that we had monthly maps in raster format, each pixel has a rainfall value in a specific month in the area in this pixel; thus, each pixel contains a time series. Before performing the clustering algorithm, we first reduced the resolution of rainfall maps by a factor of 12, where the resolution of the new monthly rainfall maps was 12 times the previous resolution (~ 254 km$^2$). Then, the clustering algorithm groups pixels based on the similarity of the shape of their rainfall time series, where each resulting group of pixels will represent a region with a similar rainfall pattern. Finally, we used a k-shape algorithm to cluster pixels by using a distance matrix constructed with a shape-based distance [48]. These clustering algorithms use the required number of groups as a parameter (i.e., the number of final regions N, so the algorithm will classify each pixel in each one of these N regions), so it is necessary to determine the optimal number of clusters. We evaluated clustering performance between N = 2 and N = 10 regions by using the silhouette (SIL) and Davies-Bouldin star (DB star) indices. These indexes evaluate the capacity of the clustering algorithm to group data by compensating the number of clusters. Thus, we selected the optimal number of clusters by searching the minimum DB star and maximum SIL indexes [49]. The result of this algorithm is a map that defines regions with similar rainfall patterns, and this algorithm was made by using the package dwtclust in R environment v. 4.1.1. [50].

**Generating incidence and precipitation data for each region.**   We calculated the average rainfall in each region by computing the mean of rainfall maps' pixels in each region. Then, to compute average incidence in each region, we first rasterized municipality-scale incidence and normalized it by the number of pixels contained in each municipality. This normalization generates raster maps of monthly incidence per pixel. Finally, we summed the incidence per pixel for all pixels contained in each region. Given that we normalized the rasterized incidence by the number of pixels per each municipality, incidence per region will not be inflated after summing incidence per pixel (i.e., National incidence by summing incidences for all municipalities and by summing incidence over all regions is the same). Nevertheless, this incidence distribution has the important assumption of spatial homogeneity within each municipality. Therefore, we did the same process for the total population for each municipality, where we got the total population from the national statistics department DANE. The outputs of this process are: *i)* Monthly precipitation for each region from 2010 to October of 2016, *ii)* Monthly reported incidence for each region for the same timespan, *iii)* Total average population for the same timespan for each region.

## Compartmental models

We formulated our compartmental models by assuming that incidence is proportional to the susceptible population ($S$) and that the parameter representing this proportionality is $\beta$: The encounter frequency with venomous snakes. We also implemented a normal random noise to this frequency, the parameter *eps*. Additionally, we assumed that each time step a proportion $\gamma$ of the envenomed population ($E$) recovers from the envenomation (Fig 1A, see S1 Text for a detailed explanation). The first model (Model 1) is our null hypothesis where climate

## A. General compartmental model

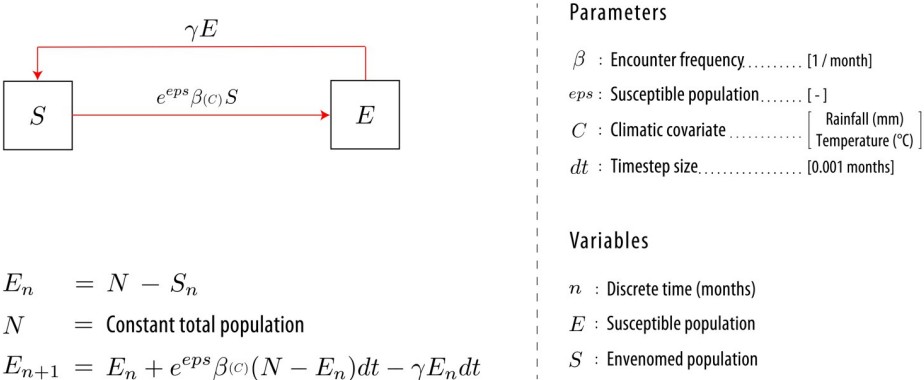

$$E_n = N - S_n$$
$$N = \text{Constant total population}$$
$$E_{n+1} = E_n + e^{eps}\beta_{(C)}(N - E_n)dt - \gamma E_n dt$$

**Parameters**

$\beta$ : Encounter frequency .......... [1 / month]

$eps$ : Susceptible population ....... [ - ]

$C$ : Climatic covariate ........... $\begin{bmatrix} \text{Rainfall (mm)} \\ \text{Temperature (°C)} \end{bmatrix}$

$dt$ : Timestep size ................ [0.001 months]

**Variables**

$n$ : Discrete time (months)

$E$ : Susceptible population

$S$ : Envenomed population

## B. Compartmental modelling scheme

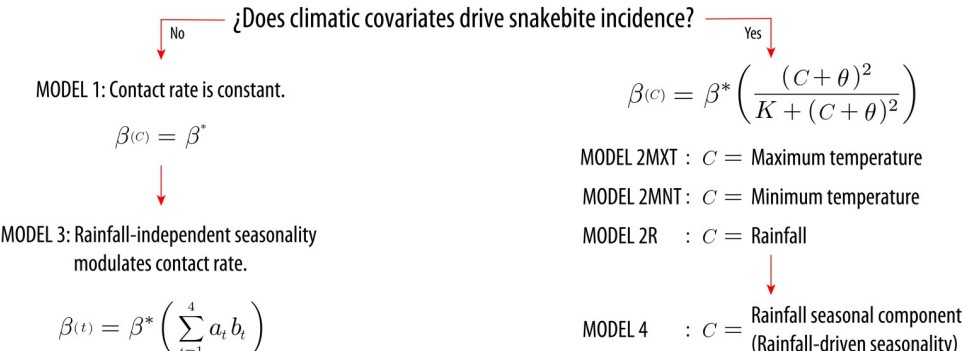

**Fig 1. General compartmental model and proposed modeling scheme. A. General compartmental model.** Our general model assumes two populations: Susceptible ($S$) and Envenomed ($E$). We defined the flow from $S$ to $E$ as the new envenoming cases per unit of time, which is the incidence. This incidence depends on a gaussian noise (*eps*), the encounter frequency parameter ($\beta$, view [27] for details), and $S$. By assuming a constant population, we defined a discrete model that only depends on $E$. **B. Compartmental modeling scheme.** We defined this scheme to test the effect of climatic covariates on snakebite incidence. First, we tested if the climatic covariate drives snakebite incidence by comparing between a Model where contact rate is constant (Model 1) with a model where this contact rate is modulated by a type III functional response (Model 2). In Model 2, we tested three climatic covariates for national data: Maximum temperature, Minimum temperature and Rainfall. Given that nationally only rainfall modulates snakebite incidence, we did the regional modeling scheme by only using rainfall variables. Then, in regions where Model 1 adjusted better data than Model 2 (Areas where incidence is not modulated by climatic covariates), we tested with Model 3 for rainfall-independent seasonality in incidence timeseries. Finally, in regions where Model 2R (Model 2 with rainfall as covariate) adjusted better data, we tested if the seasonal component of this rainfall (Rainfall-driven seasonality) is the driver behind this association by using Model 4.

covariates do not modulate this encounter frequency, but the second model does. We first used both models for national data by including as covariates for Model 2 maximum and minimum temperature and rainfall in independent models. In these second models (Model 2R: Rainfall, Model 2MXT: Maximum temperature, Model 2MNT: Minimum temperature), we established the relation between $\beta$ and climate covariates as a type III functional response (View Fig 1B).

Afterwards, we fitted the models to our defined regions to test how rainfall modulates snakebite incidence. We did not use Model 2MXT and Model 2MNT in clustered regions because we did not find any relationship between both variables and incidence in national data (View Fig 2). To test if incidence has rainfall-independent seasonality in regions where rainfall does not drive incidence (Model 1 adjusted better data than Model 2R), we proposed a third model (Model 3), which uses 4 B-splines: Here $\beta$ is a linear combination of the four B-splines. These splines are functions that make seasonal peaks on different months of the year (View Fig A in S1 Text). Finally, Model 4 tests if the seasonal component of rainfall (Rainfall-driven seasonality) is the mechanism that modulates snakebite incidence in regions where Model 2R fitted data better than Model 1. To extract the seasonal component of rainfall, we used the algorithm based on local regression described in [46]. The four models that compose our modeling scheme can be seen in Fig 1B. (View S1 Text for details).

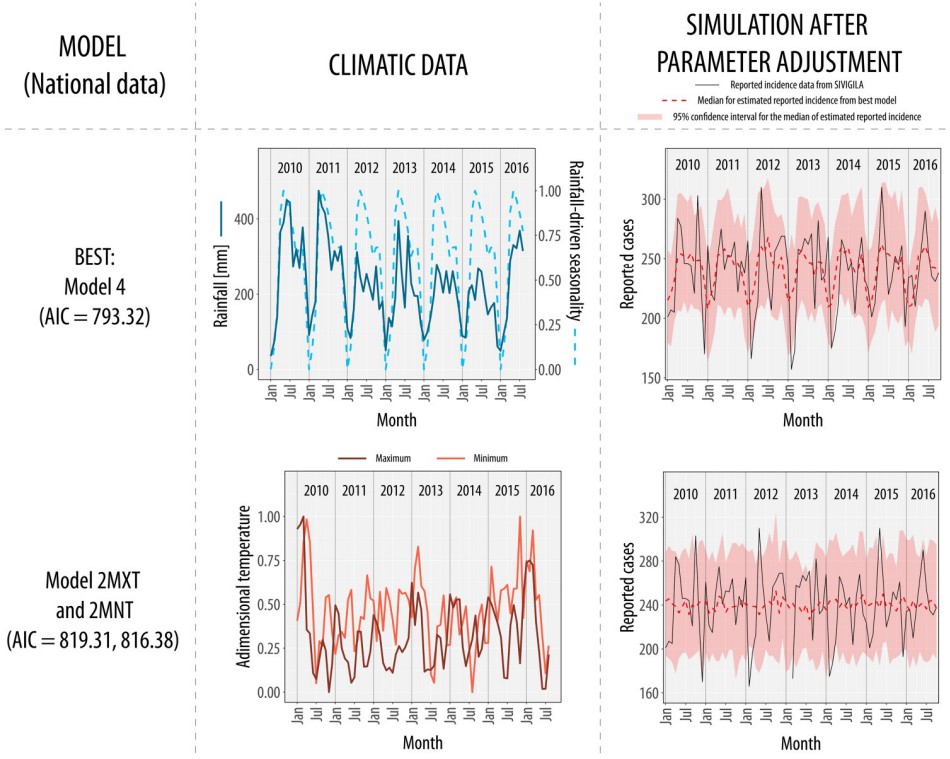

**Fig 2. Climatic data and results of the modeling scheme applied to national data.** We show the results of the best model (Model 4) and the models with temperature covariates (Model 2MXT and Model 2MNT) for national data. Climatic data shows the climatic covariates used per each model, and the simulation after parameter adjustment represents the model with the best combination of parameters that capture incidence dynamics. Note that the best model (Model 4) had lower AIC than the other models and note the difference between rainfall-driven seasonality (Dashed light-blue line) and rainfall signal (Dark blue line). In the simulations, Model 4 captures better incidence seasonality than Model 2MXT and Model 2MNT, where the median of the simulations (Dashed red line) is relatively constant.

**Fitting compartmental models to snakebite incidence data.**   We assumed the process of snakebite envenoming described by compartmental models as a partially-observed Markov process (POMP), where we can declare an observation model to estimate the error of data-reporting [44]. This observation model was a Poisson process, and the mean of this Poisson model is the monthly flow of people between S (Susceptible population) and E (Envenomed population). This flow is the reported incidence, and it is modeled as the expression shown in the arrow between S and E in Fig 1A.

Before estimating the parameters with the maximum likelihood, we detrended rainfall as we did with incidence [46], and we normalized detrended rainfall between 0 and 1. Then, we estimated the parameters that maximize the likelihood between our model and the data by iterated particle filtering, where we defined a parameter space between the biological limits of the values of the parameters of the models. We defined a random walk for each point in the parameter space, so the algorithm computes the likelihood for each combination of parameters. Finally, the algorithm converges when the likelihood reaches a "global" maximum (See S2 Text for details) [44]. To fit models, we used the 'pomp' package in R v. 4.1.1. (40).

**Best model selection.**   After computing the parameter combinations that maximize likelihood per model per region (View next section), we compared models' performance using the Akaike information criterion (AIC) to weight model capacity to represent data by its number of parameters. Thanks to this criterion, we were able to choose the best model for each region and evade overfitting caused by redundant model complexity, and we used a between-model threshold of 2 AIC units to determine significant differences [51,52]. We first selected the best model between Model 1 and Model 2R for every region, so if Model 2R outperforms Model 1, then rainfall explains the dynamics of snakebite. Then, for regions where Model 2R outperformed Model 1, we used Model 4 to test if rainfall-driven seasonality is the mechanism behind the association in both. Finally, for regions where Model 1 outperformed Model 2R (no association between rainfall and snakebite incidence), we used Model 3 to test any rainfall-independent seasonality in data.

## Results

In Colombia, between 201 and 422 envenomings are reported every month, with an average of 3659 cases per year (Min: 3135 cases in 2010, Max: 4089 cases in 2015). We found that the number of reported cases has increased with time (Pearson correlation coefficient: 0.58, p-value $< 0.05$), a result that the improving reporting system and population growth might cause rather than an increase in the actual incidence. After model selection for national data, Model 2MXT and Model 2MNT had a greater AIC value than the Model 1 (AIC for Model 1: 812.22, Model 2MXT: 819.31, Model 2MNT: 816.38); hence the temperature does not explain incidence variation in the country. On the other hand, the best model for rainfall was Model 4 (AIC: 793.323), where rainfall-driven seasonality is the driver that modulates snakebite incidence dynamics in the country (View Fig 2).

The clustering algorithm determined an optimum number of 6 regions based on the minimum David-Boulin star index value (0.74) and a silhouette index located in the first quantile of the index distribution (Silhouette index for 6 clusters: 0.46, maximum Silhouette index: 0.5 for 12 clusters) [49]. The identified clusters are shown in Fig A in S2 Text, defining the regions: *1*. South-west, *2*. Andean-Pacific, *3*. Orinoco-Amazonian piedmont, *4*. Central Amazonas, *5*. Eastern Orinoco plains, and *6*. Caribbean coast.

We fitted our compartmental modeling scheme for every region after defining these six regions with similar rainfall patterns. Interestingly, a diverse pattern of associations between rainfall and incidence throughout the 6 regions of the country was found. For example, we

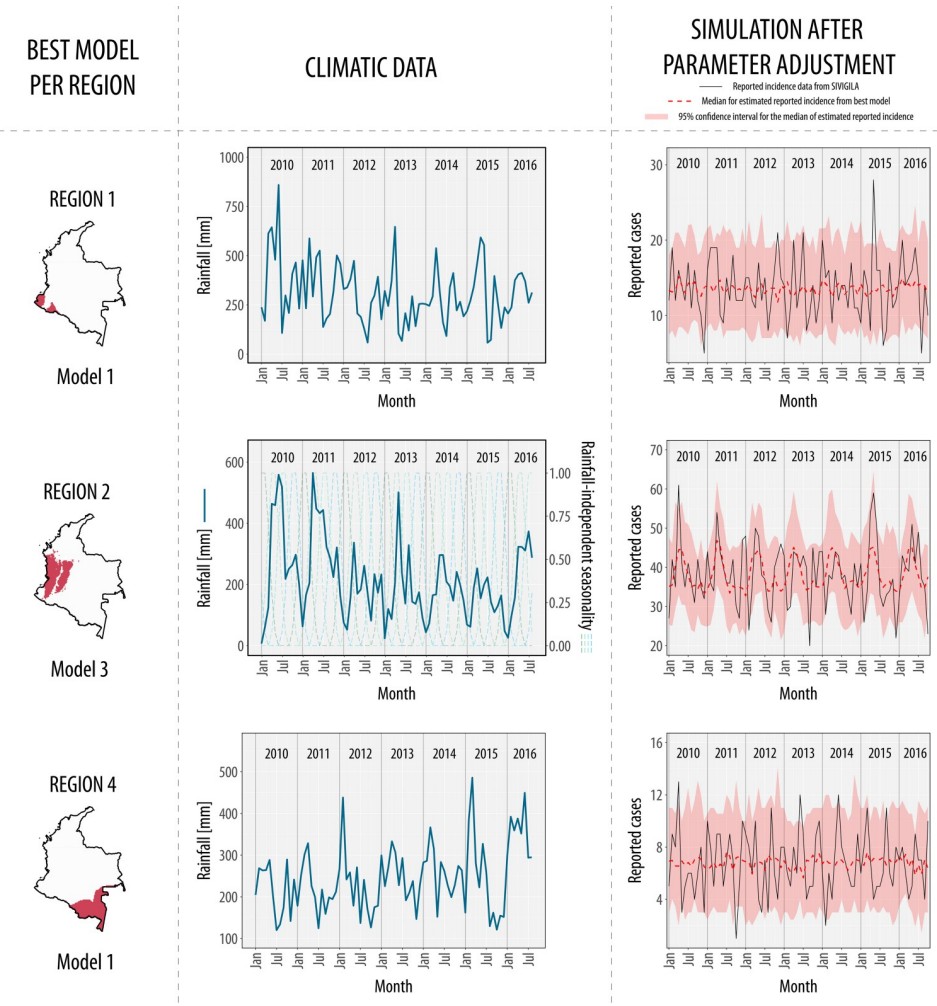

**Fig 3. Best models in regions where rainfall does not drive snakebite incidence.** Rainfall does not modulate snakebite incidence in three regions of the country, region 1, region 2, and region 4. A rainfall-independent seasonal component (Model 3) modulates snakebite incidence in region 2. This component depends on 4 B-splines (Dashed lines in climatic data plot in region2), not related to rainfall. The best model for regions 1 and 4 was Model 1, where incidence does not have seasonality and is not associated with rainfall. Note the difference between the median of the simulations (dashed red line) for region 2 and regions 1 and 4, wherein region 2 this median has seasonal peaks while in region 1 and 4 is relatively constant. Base map of national boundaries of Colombia was obtained from DIVA-GIS free spatial data (https://www.diva-gis.org/datadown).

found that rainfall does not drive snakebit incidence in three regions (region 1, 2, and 4) (View Fig 3). Regions 1 and 4 did not exhibit a seasonal component on its incidence (Model 1), while region 2 has a rainfall-independent seasonal component (Model 3). On the other hand, the incidence in regions 3, 5 and 6 is associated with rainfall-driven seasonality (Model 4), indicating that the seasonal component of rainfall signal is the one that modulates snakebite incidence dynamics (View Fig 4). The AIC values of the adjusted models for each region are shown in Table 1.

We found that the confidence intervals of the parameters determine the association between rainfall and snakebite incidence (View Table A in S2 Text). The confidence intervals for the parameter $K$ in Model 2R (model with rainfall as a covariate) are undetermined in regions where the best model was Model 1, which cancels the type III functional response

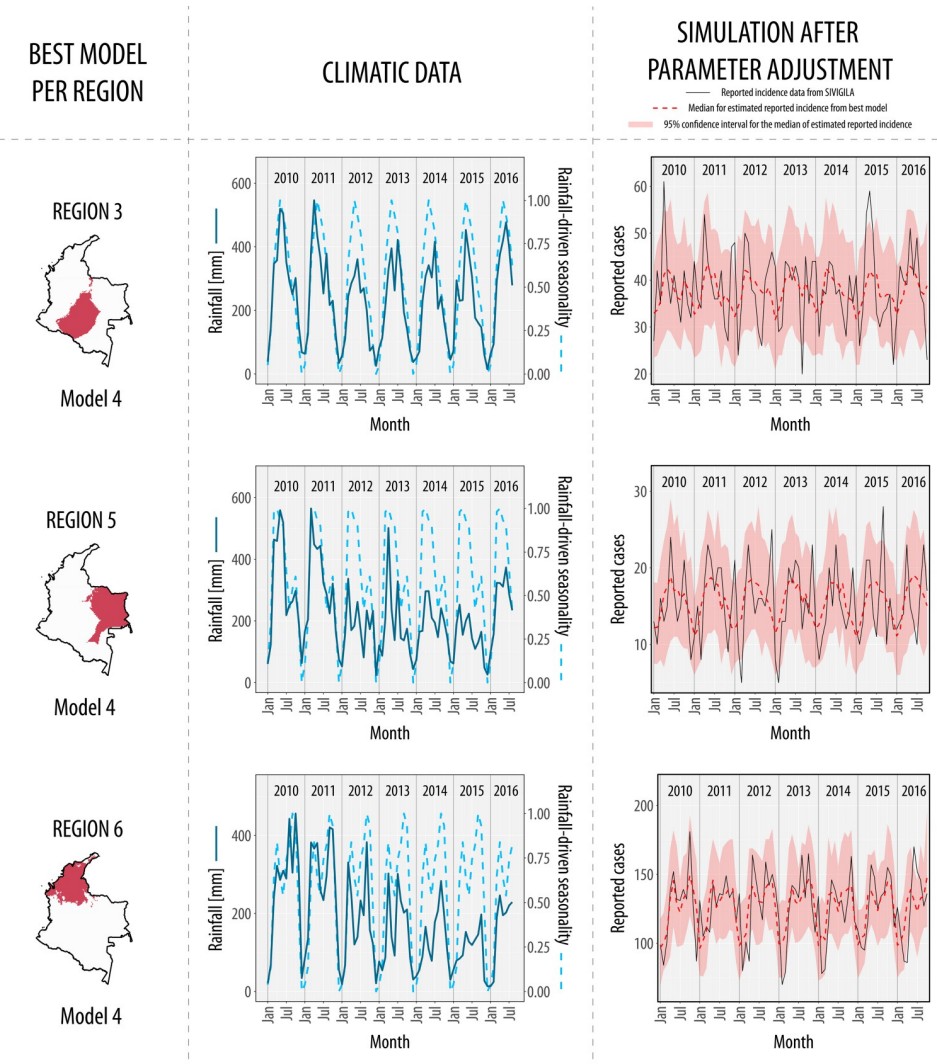

**Fig 4. Best models in regions where rainfall drives snakebite incidence.** Rainfall modulates snakebite incidence in three regions of the country, region 3, region 5, and region 6. For all regions, the best model was Model 4 that includes rainfall-driven seasonality as covariate (Dashed light-blue lines in climatic data). Note how all the medians for the simulations (dashed red lines) have seasonal peaks throughout the years, showing the association between rainfall-driven seasonality and snakebite incidence. Also, it is evident how incidence in region 6 is significantly higher than in the other regions. Base map of national boundaries of Colombia was obtained from DIVA-GIS free spatial data (https://www.diva-gis.org/datadown).

(View Table A in S2 Text). Finally, we found that in regions where rainfall-driven seasonality modulates snakebite incidence (region 3, 5 and 6), cases decrease during the dry season, and these regions have the minimum monthly rainfall (region 3: 7.02 mm, region 5: 14.88 mm, region 6: 2.73 mm) during the dry season (View Fig 4).

## Discussion

Our study explores the role of rainfall and temperature modulating snakebite incidence dynamics in different Colombian regions. We found that envenoming seasonality is significantly explained by rainfall-driven seasonality at the national level, but not by temperature. However, the association between incidence and rainfall is only present in certain regions with

**Table 1. AIC for fitted models.**

| | AIC | | | |
| | Model 1 | Model 2R | Model 3 | Model 4 |
|---|---|---|---|---|
| Region 1 | 467.52[1,2] | 467.04 | 470.18 | - |
| Region 2 | 578.6[1] | 582.58 | 564.98[2] | - |
| Region 3 | 583.12 | 580.9[1] | - | 574.69[3] |
| Region 4 | 386.52[1,2] | 390.5 | 392.7 | - |
| Region 5 | 501.94 | 488.12[1] | - | 480.9[3] |
| Region 6 | 769.68 | 752.86[1] | - | 726.84[3] |

[1] Best model between model 1 and model 2 (Does rainfall drive snakebite incidence?)

[2] Best model between model 1 and model 3 for regions where rainfall does not drive snakebite incidence (Is snakebite incidence modulated by rainfall-independent seasonality?)

[3] Best model between model 2 and model 4 for regions where rainfall drives snakebite incidence (Do rainfall-driven seasonality drives snakebite incidence?)

marked seasonal rainfall patterns. Our results suggest that national snakebite incidence has two seasonal peaks modulated by rainfall-driven seasonality, the first occurring between April and June, and the second around October (View Fig 2). This association at the national level can be explained in region 6 (Caribbean coast and Low-Magdalena region) contributing to 54% of the national cases: In this region, incidence also exhibited a significant association with rainfall-driven seasonality (View Fig 4). Therefore, inferring snakebite dynamics based only on national results should be done carefully: In region 2 (Andean and pacific regions), the peak of envenomings occurs during the beginning of the year, and they are modulated by rainfall-independent seasonality, in contrast with the national trend (View Fig 3). Contrary to other studies about snakebite seasonality that have used national incidence data [15,16], our study compares snakebite risk between different regions by analyzing hypotheses represented by compartmental models: National-aggregated analyses can largely neglect the geographical heterogeneity in the association between snakebite and temporal drivers, as it was shown in [40].

## Spatial distribution of snakebite risk in Colombia

By looking at the null model (No rainfall, temperature, nor rainfall-independent seasonality as covariates, Model 1), the parameter $\beta$ represents a "constant" snakebite risk or encounter frequency with venomous snakes as described in [27]. Therefore, the likelihood profile over this parameter (View Table A in S2 Text) and its confidence intervals (View Fig B in S2 Text) can be used to compare risk between regions: The regions with the highest risk (region 1, 3, 4, and 5) have the presence of *B. atrox* [33,53], while regions with *B. asper* (regions 2 and 6) are the regions with the lowest risk. This effect can be caused by ecological differences between both species, which makes *B. atrox* more dangerous or abundant than *B.asper*, or by economic or sociological differences that can increase the exposure to snakebite of inhabitants of these regions. Sadly, biological information about venomous species in the country is scarce, so it is difficult to determine the cause of these high-risk regions [11,30,33,53,54].

## Temporal patterns of snakebite incidence in Colombia

We found that regions with the highest coefficient of variation for rainfall, which determines how seasonal is the temporal pattern of precipitation (regions 3, 5, and 6, View Table B in S2 Text), have their reported incidence modulated by rainfall-driven seasonality, where incidence decreases during dry seasons (Model 2R explained snakebite incidence better than Model 1). Thus, a strong seasonality on rainfall, characterized by marked dry and rainy seasons,

determines the association between snakebite incidence and rainfall in Colombia. Therefore, we propose that rainfall acts as a limiting resource in snakebite dynamics in Colombia: The association between snakebite incidence and rainfall in Colombia is mediated via marked dry seasons, which cause a decrease in venomous snakes' activity, abundance, and finally, snakebite risk [37,55].

Another important time series seasonality measurement is the strength of seasonality, which compares the variance in the noise component of the time series with the variance in the seasonal component [33]. Rainfall seasonal strength was considerably lower in regions where we did not find an association between precipitation and incidence; consequently, the rainfall signal in these places is noisy and does not have a clear temporal periodic trend (View Table B in S2 Text and Fig C in S2 Text). It is important to clarify that region 1 has a part located in the Pacific versant, and another is located in the Amazonian versant, which are two distinct ecological regions with different diversity of venomous snakes, but a similar rainfall pattern [33,53,56]. Thus, these differences in venomous snakes' diversity and ecology can explain the high noise over incidence time-series in this place. We recommend investigating more deeply the association between rainfall and incidence in this region, where data from Ecuador, which is the neighboring country to this area and has the same species composition in the Pacific and Amazonian versant [53], can help to clarify the dynamics of snakebite incidence in this region.

The mechanisms behind the association between rainfall and snakebite incidence patterns are still unclear. However, we want to propose three not mutually exclusive hypotheses to explain this binding: *i)* Several neotropical snakes, including some species of the genus *Bothrops*, have their reproductive cycle related to precipitation pattern: Gravid females give birth to neonates at the beginning of the rainy season, thus increasing the abundance of venomous snakes and the probability of encounter between humans and venomous snakes [10,11]. For example, in Costa Rica, the reproductive cycle of *Bothrops asper* is known, and the seasonality of snakebite incidence is driven by the population dynamics of this species [11,15]. These dynamics can explain the rainfall-independent seasonality of incidence found in region 2, and the association between incidence and rainfall-driven seasonality found in region 6. Nevertheless, *Bothrops asper* populations in Colombia are genetically different from populations in Costa Rica, so population dynamics between both populations may vary [57]. In addition, it is known that in Brazilian Amazonas, the reproductive cycle of *Bothrops atrox* is not seasonal, where births occur during most of the year [12]. This can explain why snakebite incidence is not associated with precipitation or rainfall-independent seasonality in region 4 (central Amazonas), but for regions 3 and 5, where *Bothrops atrox* is also present, the incidence is modulated by rainfall-driven seasonality. *ii)* Precipitation can affect the ecology of venomous snakes, either by causing floods which decrease the not flooded area that snakes and humans share, therefore raising the contacts between both populations, or by increasing ecosystem productivity: More prey will be available, so snakes could be more active thus increasing snakebite risk [11]. *iii)* During rainy seasons, agricultural and cattle productivity increases, causing an increase in the number of farmer workers at risk of encountering a venomous snake [58]. Given that ecological information about venomous snakes is non-existent in Colombia, fieldwork must be done to determine how these three rainfall-related events affect snakebite incidence temporal patterns and determine risky seasons.

## Final remarks

Our modeling framework can describe the dynamics of snakebite incidence in Colombia. We believe this modeling framework can be used easily in other countries to monitor snakebite

incidence and to improve disease management under changing environments. It is crucial to account for regions with different precipitation patterns to determine spatially heterogeneous snakebite dynamics, which was also addressed recently in [40] by statistical modeling. These results demonstrate how countries affected by snakebite can determine in which regions and in which time they need more antivenom, and how to distribute this scarce resource more accurately. This strategic distribution can help decreasing disease burden because most places affected by snakebite have a deficit in antivenom coverage [4,5,7,19]. In addition, determining the ecological mechanism behind model-estimated snakebite temporal patterns can help to develop prevention strategies to decrease the burden of this severe NTD.

We used a compartmental model to understand the temporal patterns of snakebite incidence by using as explanatory variables rainfall, temperature, and a rainfall-independent seasonal component. Given that these models are robust and their structure is based on the available knowledge behind the causes of the disease, several modifications, hypotheses and extrapolations can be tested: We encourage researchers to use this modeling scheme to understand snakebite epidemiology in other countries. For example, in contrast with statistical modeling, in our compartmental models, the estimated value for the parameters represents characteristics of the ecology of the disease (i.e., encounter frequency $\beta^*$ has units of the number of encounters with venomous snakes that ends in a snakebite per unit of time). Even so, our approach is a first step in the development of a better understanding and prediction of snakebite epidemiology by mathematical modeling: Epidemiological models are more refined models that explicitly use information on snake abundance instead of the mentioned environmental proxies, which can deal with the correlation-causality ambiguity that is still present in our approach.

Epidemiological models have been used in several neglected tropical diseases caused by zoonosis, where the basic biology of the hosts involved in disease transmission is known and modelled [59–62]. Thanks to this synergy between mathematics and biology, more specific control programs have been proposed to control the disease burden over an infected population [29,63–65]. Thus, as it has been proposed before, the role that the biology of venomous snakes plays behind snakebite epidemiology is important, but it is still neglected: We want to encourage the study of the natural history of venomous snakes to fill this enormous vacuum of information that limits the understanding of snakebite, and which can be used to generate more robust epidemiological models by including venomous snakes population dynamics. Finally, an interdisciplinary approach must be undertaken to decrease the high burden of this NTD, and to help contribute to the WHO target to reduce snakebite fatality by 30% by the end of 2050 [66].

## Supporting information

**S1 Text. Compartmental models and B-splines.** This section contains the development and explanation of the models. We used four models in our modeling framework to determine the association between climatic covariates and snakebite incidence, which are explained in Fig A. (DOCX)

**S2 Text. Settings for parameter estimation and model selection, clustered regions with similar rainfall patterns, confidence intervals after parameter adjustment, and seasonal parameters for rainfall over study areas.** We used a Poisson process to model data gathering, and then we used a particle filtering algorithm in two steps to adjust our modeling scheme to the data. This process is explained here. Fig A shows the results of the regions with similar rainfall patterns after clustering algorithm. Table A shows the confidence intervals for all parameters in all models after particle filtering algorithm. Fig B compares the adjusted

encounter frequency for MODEL 1 between all regions, showing that regions where *Bothrops atrox* is distributed share the highest values. Table B shows seasonal parameters for rainfall in our defined regions and Fig C contains the seasonal distribution of rainfall over clusterized regions.
(DOCX)

## Acknowledgments

We thank to Juan Daniel Umaña, Daniela Angarita, Norma Forero, Gabriela Navas and Carlos Cruz for the teamwork doing preliminary statistical analyses, Alejandro Feged for his support at the beginning of the project, and Mahmood Sasa and Camila Renjifo for their support during the study analysis.

## Author Contributions

**Conceptualization:** Carlos Bravo-Vega, Juan Manuel Cordovez.

**Data curation:** Carlos Bravo-Vega, Mauricio Santos-Vega.

**Formal analysis:** Carlos Bravo-Vega, Mauricio Santos-Vega.

**Funding acquisition:** Juan Manuel Cordovez.

**Supervision:** Juan Manuel Cordovez.

**Validation:** Carlos Bravo-Vega, Mauricio Santos-Vega.

**Visualization:** Carlos Bravo-Vega.

**Writing – original draft:** Carlos Bravo-Vega.

**Writing – review & editing:** Carlos Bravo-Vega, Mauricio Santos-Vega, Juan Manuel Cordovez.

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
