## [Decision Letter · Decision Letter 0]

14 Oct 2021

Dear Dr Bravo-Vega,

Thank you very much for submitting your manuscript "Disentangling snakebite dynamics in Colombia: How does rainfall drive snakebite temporal patterns?" for consideration at PLOS Neglected Tropical Diseases. As with all papers reviewed by the journal, your manuscript was reviewed by members of the editorial board and by several independent reviewers. In light of the reviews (below this email), we would like to invite the re-submission of a significantly-revised version that takes into account the reviewers' comments. 

As you will note, all reviewers found substantial novelty and noted the importance of the work. However, issues around methodology and ease of understanding (especially of figures) were made by several reviewers. Please take careful note of the reviewers comments when preparing for re-submission. 

We cannot make any decision about publication until we have seen the revised manuscript and your response to the reviewers' comments. Your revised manuscript is also likely to be sent to reviewers for further evaluation.

Sincerely,

Stuart Robert Ainsworth

Associate Editor

José María Gutiérrez

Deputy Editor

Reviewer's Responses to Questions

**Key Review Criteria Required for Acceptance?**

**Methods**

-Are the objectives of the study clearly articulated with a clear testable hypothesis stated?

-Is the study design appropriate to address the stated objectives?

-Is the population clearly described and appropriate for the hypothesis being tested?

-Is the sample size sufficient to ensure adequate power to address the hypothesis being tested?

-Were correct statistical analysis used to support conclusions?

-Are there concerns about ethical or regulatory requirements being met?

Reviewer #1: I only have two problems with methods. First is that, while methods are robust and novel in the field, I do not think that resulting models are epidemiological because they statistically estimate one parameter, labelled frequency of contact, as a function of rainfall. The authors try and convince the reader that because the estimated parameter is the frequency of contact, their models do not suffer from the correlation-causation ambiguity, although in essence that parameter is not different from previous estimates of incidence as a function of the environment. Therefore I suggest that such a description of their results should be avoided. I think then that the authors should seek to convince us that they have found a way to separate different sources of spatio-temporal variability of snakebite risk, which is on its own a very relevant development.

The second issue is related to data aggregation, in the conversion of municipalities from vector to raster, because vector data may have polygons larger than pixels, so when a polygon is converted to smaller pixels, all the pixels within a particular polygon will have the same population value. When these values are aggregated the resulting pixels will indicate a population size larger than it sould be. Currently there is no information on whether this taken care of, or if it was considered, or if it actually is not a relevant issue and why. 

Another minor issue I have with methods is related to writing. There is a tendency to use jargon in places where we need to understand what authors did, for instance line 182 is very difficult to make sense, not only because it contains very technical terms, but because there are quantities mentioned of which there are no details given. Also, there is a lot of room for confusion in the use of rainfall and seasonality, because in occasions rainfall means seasonality, and viceversa.

Reviewer #2: Major Revision

Reviewer #3: The study objectives are clearly articulated, the study design is adequate, and the statistical analyses are appropriate. I have no ethical concerns.

**Results**

-Does the analysis presented match the analysis plan?

-Are the results clearly and completely presented?

-Are the figures (Tables, Images) of sufficient quality for clarity?

Reviewer #1: Figures are all of very good quality, but captions should contain more succinct and understandable information. For instance, very few of us know how to interpret the "cross-wavelet" subfigure in Fig 1, so I don't know what's its relevance or what aspect of the data does it explain, or describes, or which one of the phenomena in question it refers to. What do we learn about snakebites from this particular figure?

Reviewer #2: Figures are difficult to understand, please increase the clarity and add explanation. Especially figure 5, need to provide details.

Reviewer #3: The results, figures and tables matches the analysis plan and I have no concerns as to their quality.

**Conclusions**

-Are the conclusions supported by the data presented?

-Are the limitations of analysis clearly described?

-Do the authors discuss how these data can be helpful to advance our understanding of the topic under study?

-Is public health relevance addressed?

Reviewer #1: The only major concern I have is with the narrative of their epidemiological models no having the causality problem one encounters with statistical models. This issue reappears in an important way in the conclusions.

Reviewer #2: Accept

Reviewer #3: The conclusions are adequate and supported by the analysis, besides some minor concerns I've outined in my comments.

**Editorial and Data Presentation Modifications?**

Reviewer #1: There are parts of the manuscript that need rephrasing, especially in methods.

Introduction

Line 73. I’d say, it’s strictly due to extrapolation and the process complexity that such models hide.

Line 74. disease spread, not diseases’ spread.

Line 75. Models in NTDs have been mainly used for endemic dynamics, not spread as in epidemic invasion, so i’d rephrase the previous sentence to reflect that characteristic of NTDs, it’s transmission rather than spread.

Line 89. This group is vivparous... each female gives birth to several fully developed newborns.

Line 100. Colombia is an ideal setting...

Materials and methods

Line 110. That accounts for the role of climate. A model by itself, I beg to disagree, cannot “understand”.

Line 114. municipality-reported incidence, remove “the” before incidence.

Line 132. What does “ones” refer to? Does it mean that you added beta-estimates from both models? Also, I don’t really follow the sentence “was only used on clusters...” does it mean that you only used model 4 on certain areas where the rainfall signal is stronger?

Line 140. Please define E before first mention. A general description of the model as is SIR would be useful. I also think your paper will benefit a lot from having a diagram that shows geometrically the difference between rainfall models.

Line 150. Suggest: parameter combinations, instead of “parameters combination”

Line 152. Suggest: if model likelihoods were significantly different between each other

Line 158. Suggest: a threshold of significance between likelihood differences.

Line 162. Suggest: To fit models we used the pomp R package.

Line 182. By a factor of 12 what? Did you aggregate 12x12 pixels? Pixels around a 12 km radius?

Line 183. I think it’s easier to describe what the clustering procedure did, rather than list the names of the technics. While not being familiar with the methods, it’s very difficult to understand what you did.

Line 183 Between 2 and 10 what? In general this paragraph is very difficult to understand, I get the feeling that it is about summarising precipitation data in a way that best explains snakebite incidence data, but since it mainly contains very technical terms it’s hard to make sense out of it.

Lines 190-192. Are municipality and area synonims here? How do these relate to “clusters”. I get the impression that you’re describing two ways of aggregating data, but it’s not clear at all. 

Results

Lines 201-205. This should be in methods section.

Line 206. Does “estimated seasonality” mean something like seasonal periods? So 6 months means two annual incidence “peaks” whilst 12 monts means one seasonal peak? Please clarify, because it is the whole point of the paper and should be spelled as clearly as possible in terms that non-modellers can understand.

Figure 1. I think the cross-wavelet figure and cross correlation function need further explanation. How do these sub-figures related to the biological phenomenon of snakebite? How can these subfigures be interpreted and what do they reflect about the de-trended data? Are they relevant to your main results?

Line 224. Figure 2 caption: models fitted to each region.

Line 226. Figure 2 caption: indices instead of indexes.

Line 237. so the person becomes... No need to assume any particular gender!

Figure 2 caption. I think the use of seasonality and rainfall is extremely confusing, are you saying that seasonality is a process independent of rainfall? This should be made clearerer and probably using different terms such as rainfall-driven seasonality vs rainfall-independent seasonality. I now see how the use of these terms relate to the model (4) but it can be made less confusing from the start if you use more informative names.

Lines 250-253. This is just repetition from methods, so I think it can be removed unless you want to do a shorter recap.

Lines 256-259. The use of the term national-aggregated data adds another layer of confusion here. I don’t know if it refers to the entire country’s incidence or to the particular clusters where model 4 was the “best”. Also, I think you should say nationally-aggregated data, unless I’m missing something.

Lines 278-280. We found that in three regions .... rainfall does not explain incidence.

Line 281. in region 5, there is a stronger relationship between snakebite and rainfall…

Figure 3 caption (lines 287-303). Please fix the same aspect about other seasonality sources commented above.

Line 299. precipitation drives snakebite strongly (at the end)

Line 303. This sounds like only region 5 has seasonal snakebites, which is why you need to make a clearer use of the terms rainfall and seasonality.

Lines 304-312. This paragraph needs a topic sentence.

Discussion

Line 314. … the role of rainfall on snakebite seasonality.. (this needs more consistent use and clarification).

Line 342. Again, seasonality…

Line 344. By a larger rainfall pattern do you mean that the difference between wet and dry seasons is very large? 

Lines 347-349 Suggestion: incidence and rainfall in Colombia is mediated via snakes’ activity… (delete what’s in between sentences)

Line 395. Same as per my comment above.

Line 414. “undertaken”, instead of “done”.

Reviewer #2: Accept

Reviewer #3: There are many minor grammatical and wording issues that I would like to see addressed as outlined in the word document containing my comments.

**Summary and General Comments**

Reviewer #1: The strengths of the study are that it uses statistically robust methods implemented in a very novel way to identify models that better explain data in different geographical areas to characterise sources of temporal risk variability. I believe that this alone is a sufficiently strong development to recommend publication, given that the appropriate changes are made.

With regards to weaknesses, methods are statistical, as I mentioned, so ascribing causality to the relationships via parameter estimates, as authors claim, is still difficult.

Reviewer #2: Disentangling snakebite dynamics in Colombia: How does rainfall drive snakebite temporal patterns?

I must congratulate all the authors for carrying out this work. This is an important work in the field of snakebites. However, I have a few comments. Please refer my comments.

Reviewer #3: The manuscript is an important contribution to the field and will benefit snakebite management. Further comments can be found in my detailed comments in the attached word document.

PLOS authors have the option to publish the peer review history of their article (what does this mean?). If published, this will include your full peer review and any attached files.

Reviewer #1: Yes: Gerardo Martín

Reviewer #2: No

Reviewer #3: No
---

## [Decision Letter · Decision Letter 1]

1 Feb 2022

Dear Dr Bravo-Vega,

Thank you very much for submitting your manuscript "Disentangling snakebite dynamics in Colombia: How does rainfall and temperature drive snakebite temporal patterns?" for consideration at PLOS Neglected Tropical Diseases. As with all papers reviewed by the journal, your manuscript was reviewed by members of the editorial board and by several independent reviewers.

The reviewers appreciated the attention to an important topic and have noted its substantial improvement in quality.

You will note reviewer 1 still has some minor points of clarification they would like attention to. Based on the reviews, we are likely to accept this manuscript for publication, providing that you modify the manuscript according to the review recommendations. 

Sincerely,

Stuart Robert Ainsworth

Associate Editor

José María Gutiérrez

Deputy Editor

Reviewer's Responses to Questions

**Key Review Criteria Required for Acceptance?**

**Methods**

-Are the objectives of the study clearly articulated with a clear testable hypothesis stated?

-Is the study design appropriate to address the stated objectives?

-Is the population clearly described and appropriate for the hypothesis being tested?

-Is the sample size sufficient to ensure adequate power to address the hypothesis being tested?

-Were correct statistical analysis used to support conclusions?

-Are there concerns about ethical or regulatory requirements being met?

Reviewer #1: (No Response)

Reviewer #3: The new analysis is clearly described and suitable for the purpose of the study.

**Results**

-Does the analysis presented match the analysis plan?

-Are the results clearly and completely presented?

-Are the figures (Tables, Images) of sufficient quality for clarity?

Reviewer #1: (No Response)

Reviewer #3: The presented results match the analysis plan and are clearly presented.

**Conclusions**

-Are the conclusions supported by the data presented?

-Are the limitations of analysis clearly described?

-Do the authors discuss how these data can be helpful to advance our understanding of the topic under study?

-Is public health relevance addressed?

Reviewer #1: (No Response)

Reviewer #3: The conclusions are supported by the results and discussed results are highly relevant to public health.

**Editorial and Data Presentation Modifications?**

Reviewer #1: Overall I am very impressed by how the paper has improved. I only have one minor general comment and related to the same major problem I pointed out before: this study's main limitation has to be acknowledged from the beginning; the introduction suggests that compartmental models are infalible in comparison with empyrical ones (lines 84-96, ), whereas the "frequency" of contact estimated in this study is empyrical which means that it has the same limitations of previous studies that estimate incidence. Therefore the introduction should not sell the idea that these models are mechanistic in the same sense that epidemiological models are. In relation to the suggestion made in the abstract about model extrapolation, I think it's extremely risky because overall ecology would not hold outside the Colombian setting, human-snake contact rates also reflect socio-economic aspects of local populations in relation to local snake species and in this case hospital admissions.

Here are some other minor editorial suggestions.

Line 112 "statistical" instead of "statistic"

Line 137 Sistema nacional de vigilancia nacional, which is correct, Sistema Nacional de Vigilancia or Sistema de Vigilancia Nacional?

Line 167 Should it be indices instead of indexes?

Line 214 We fitted the models

Line 215 We did not use

Line 222 Model XX Fitted data better than model YY

Line 250 After computing the parameter (singular) combinations

Line 254 Suggestion: a between-model threshold of 2 AIC units to determine ...

Line 264 Is it possible that there is also a role of population growth? The difference between population growth and reports should the attributable effectively to improved reporting, shouldn't it?

Line 329 Can be explained in region 6 ....

Line 364 Rainfall seasonal strength ...

Line 366 And does not have a clear temporal, periodic trend?

Line 420 Suggestion : ... more robust models that explicitly use information on snake abundance instead of the mentioned environmental proxies ...

Reviewer #3: The new version of the anuscript is clear and I have no further suggestions.

**Summary and General Comments**

Reviewer #1: This paper provides novel methodological alternatives to analyse snakebite incidence data and identify different categories or risk areas based on ecological and environmental characteristics. As I mention before study limitations have to be acknowledged: the mised empyrical-mechanistic nature of the analyses and inability to extrapolate in light of the partially empyrical nature of the analyses.

Reviewer #3: Overall, the manuscript is an important contribution to the field and provides many novel insigths into snakbit dynamics.

PLOS authors have the option to publish the peer review history of their article (what does this mean?). If published, this will include your full peer review and any attached files.

Reviewer #1: Yes: Gerardo Martin

Reviewer #3: No

Figure Files:

Data Requirements:

Reproducibility:

References

---

## [Decision Letter · Decision Letter 2]

21 Feb 2022

Dear Dr Bravo-Vega,

We are pleased to inform you that your manuscript 'Disentangling snakebite dynamics in Colombia: How does rainfall and temperature drive snakebite temporal patterns?' has been provisionally accepted for publication in PLOS Neglected Tropical Diseases.

Best regards,

Stuart Robert Ainsworth

Associate Editor

José María Gutiérrez

Deputy Editor

Reviewer's Responses to Questions

**Key Review Criteria Required for Acceptance?**

**Methods**

-Are the objectives of the study clearly articulated with a clear testable hypothesis stated?

-Is the study design appropriate to address the stated objectives?

-Is the population clearly described and appropriate for the hypothesis being tested?

-Is the sample size sufficient to ensure adequate power to address the hypothesis being tested?

-Were correct statistical analysis used to support conclusions?

-Are there concerns about ethical or regulatory requirements being met?

Reviewer #1: (No Response)

**Results**

-Does the analysis presented match the analysis plan?

-Are the results clearly and completely presented?

-Are the figures (Tables, Images) of sufficient quality for clarity?

Reviewer #1: (No Response)

**Conclusions**

-Are the conclusions supported by the data presented?

-Are the limitations of analysis clearly described?

-Do the authors discuss how these data can be helpful to advance our understanding of the topic under study?

-Is public health relevance addressed?

Reviewer #1: (No Response)

**Editorial and Data Presentation Modifications?**

Reviewer #1: (No Response)

**Summary and General Comments**

Reviewer #1: I don't have any further comments and will be glad to see this paper published.

PLOS authors have the option to publish the peer review history of their article (what does this mean?). If published, this will include your full peer review and any attached files.

Reviewer #1: **Yes: **Gerardo Martín

---

## [Editor Report · Acceptance letter]

9 Mar 2022

Dear Bravo-Vega,

We are delighted to inform you that your manuscript, "Disentangling snakebite dynamics in Colombia: How does rainfall and temperature drive snakebite temporal patterns?," has been formally accepted for publication in PLOS Neglected Tropical Diseases.

Best regards,

Shaden Kamhawi

co-Editor-in-Chief

Paul Brindley

co-Editor-in-Chief
